# Development of Lateral Flow Immunochromatographic Test for Rapid Detection of SARS-CoV-2 Virus Antigens in Clinical Specimens

**DOI:** 10.3390/nano12142477

**Published:** 2022-07-19

**Authors:** Rafik Hamed Sayed, Mohamed Samy Abousenna, Shaimaa Abdelall Elsaady, Rafik Soliman, Mohamed Ahmed Saad

**Affiliations:** 1Central Laboratory for Evaluation of Veterinary Biologics, Agricultural Research Center, P.O. Box 131, Cairo 11381, Egypt; shaimaaabdelall@gmail.com; 2Department of Microbiology, Faculty of Veterinary medicine, Cairo University, Giza 12211, Egypt; rafiksoliman108@hotmail.com; 3Veterinary Serum and Vaccine Research Institute, Agricultural Research Center, P.O. Box 131, Cairo 11381, Egypt; saad940@yahoo.com

**Keywords:** SARS-CoV-2, COVID-19, lateral flow assay, antigen testing, sensitivity and specificity, diagnostic testing, immuno-chromatographic, gold nanoparticles

## Abstract

In the presented study, we developed a nanogold lateral glow immunoassay-based technique (LFI-COVID-19 antigen test) for the detection of SARS-CoV-2 nucleocapsid proteins; the developed LFI-COVID-19 Ag test has been tested for limit of detection (LOD), cross-reactivity and interfering substances, and performance. It was found that the performance of the developed LFI-COVID-19 antigen test when it was evaluated by RT-qPCR indicated 95, 98, and 97% for sensitivity, specificity and accuracy, respectively. This complies with the WHO guidelines. It was concluded that the developed LFI-COVID-19 antigen test is a point of care and an alternative approach to current laboratory methods, especially RT-qPCR. It provides an easy, rapid (within 20 min), and on-site diagnostic tool for COVID-19 infection, and it is a cheap test if it is manufactured on a large scale for commercial use.

## 1. Introduction

The coronavirus disease 2019 (COVID-19) was caused by the novel betacoronavirus, namely severe acute respiratory syndrome coronavirus 2 (SARS-CoV-2) [1,2]. It was detected in 2019 and it has spread to most countries worldwide, with noticed high transmission and morbidity rates. It belongs to the family Coronaviridae in the order Nidovirales, genera Betacoronavirus. It had been isolated in Wuhan, and it was hypothesized that the SARS-CoV-2 is of bat origin [3,4].

COVID-19 has been recorded in most countries of the world, with noticed high transmission and morbidity rates that triggered the alarm for public health to respond to emerging diseases [5]. The World Health Organization (WHO) declared the COVID-19 outbreak a worldwide pandemic on 11 March 2020. As of 22 November 2021, the World Health Organization recorded more than 5 million deaths and more than 256 million confirmed cases worldwide [6].

A comprehensive strategy, including diagnostics, extensive surveillance, precautionary measures, antiviral drugs, and vaccine development, is urgently needed to combat COVID-19 [7].

Extensive surveillance promotes early diagnosis and rapid response to combat COVID-19 efficiently. Diagnostic tests have a major role in tracking the disease spread. The real-time reverse transcription polymerase chain reaction (RT-qPCR) has been used as a priority diagnostic tool in early infection by different authorities in most countries around the world [8]. However, this method has some shortcomings in the rapid screening of asymptomatic carriers, early-stage patients and the infected patients with recent variants because of its high false-negative rates. In addition, RT-qPCR requires a long time for testing, advanced laboratory facilities and professional personnel. Conversely, simple and cheap antigen-based rapid tests in the format of lateral flow assays (LFA) have been developed to address these issues for SARS-CoV-2 diagnosis [9,10,11]. Moreover, the lateral flow SARS-CoV-2 antigen detection devices (LFDs) are an alternative point of care solution that can quickly turn around a result in less than 30 min at the bedside [12,13].

Lateral flow immunoassays (LFIAs) or lateral flow assays (LFAs) are very attractive for home testing and have been considered one of the most popular point-of-care testing techniques, since they are affordable, rapid, cheap, flexible, scalable and demonstrate wide adaptability [10,14].

The present study has been conducted to develop a lateral flow immunoassay for the detection of SARS-CoV-2 nucleocapsid and spike proteins. The study also set out to identify the sensitivity and specificity of the developed lateral flow immunoassay in the detection of SARS-CoV-2 virus in clinical specimens from patients suspected of COVID-19 virus infection as compared to RT-qPCR.

## 2. Materials and Methods

### 2.1. Nucleocapsid Protein (NP) and Spike Protein (S) of SARS- CoV-2

The SARS-CoV-2 spike (S) protein (Cat: 40591-v08H) and the nucleocapsid (NP) protein (Cat: 40588-v07E) were purchased from Sinobiological, Chesterbrook, PA, USA.

### 2.2. Preparation of Monospecific Antibodies against NP in Rabbits

The nucleocapsid protein (NP) solution was mixed with complete Freund’s adjuvant volume/volume percent solution. Twenty-five male rabbits were inoculated with 0.1 mg/dose of the emulsion, intradermally. Booster doses of the nucleocapsid protein (NP) and incomplete Freund’s adjuvant mixture were inoculated (0.1 mg/dose, subcutaneously) in the previously immunized rabbits at the 2nd, 4th, 6th and 8th week intervals. Serum samples were collected at 5 days post last inoculation, and tested for SARS-CoV-2- NP-specific antibodies using an agar gel precipitation test. If the serum sample contains antibodies to antigens of NP, they bind together, forming an interlaced antigen–antibody complex that precipitates in the agar. The precipitate is visible to the unaided eye as a thin white line [15].

### 2.3. Preparation of Monospecific Antibodies against S Protein in Goats

Spike protein (S) was mixed with complete Freund’s adjuvant volume/volume percent solution. Five male goats in total were inoculated with 0.1 mg/kg dose of the mixture intradermally. Booster doses of the Spike protein (S) and incomplete Freund’s adjuvant mixture were inoculated (0.1 mg/Kg dose, subcutaneously) in the previously immunized goats at the 2nd, 4th, 6th and 8th week intervals. Five days post the last inoculation, the serum samples containing goat SARS-CoV-2- S-specific antibodies were collected and tested with an agar gel precipitation test [15].

### 2.4. Purification of Immunoglobulin of Rabbit and Goat Antibodies Using Caprylic Acid

The collected serum samples from rabbits immunized with SARS-CoV-2 nucleocapsid (NP) proteins were pooled and centrifuged at 12,000 rpm for 30 min and the pellet was discarded. The collected serum was diluted 1:3 using sodium acetate buffer (0.06 M, pH 4.6) in a beaker with a magnetic bar and stirred. Two mL of caprylic acid were added dropwise to the diluted rabbit serum after 30 min stirring at room temperature, then it was centrifuged for 30 min at 12,000 rpm. The supernatant was harvested, and dialyzed overnight with PBS buffer at 4 °C. Two or three buffer changes must be carried out to remove the excessed caprylic acid. The purified immunoglobulin concentration was detected using a spectrophotometer. The same procedure was conducted for the purification of the IgG from goat serum immunized with SARS-CoV-2 spike protein [16].

### 2.5. Preparation of Nanogold Particles of 40 nm Diameter Size

Boiling 50 mL of ultra-pure water with vigorous stirring was carried out using a hot plate stirrer and by adding sodium citrate 0.01% (w/v). HAuCl_4_ 1% 1 mL was added to the solution. When the solution color became red, this was referred to as the gold nanoparticles formulation. Then, sodium azide 0.02% (w/v) was utilized. The solution was kept for cooling and the spectrophotometer was used to confirm the diameter of the obtained nanogold particles within the range 400–600 nm [17].

### 2.6. Conjugation of Nanogold Particles with Purified Rabbit IgG Specific to SARS-CoV-2 Nucleoprotein

The pH of the nanogold particles was adjusted to 8.5 using 0.02 M K_2_CO_3_. With gentle stirring, the rabbit-purified IgG 1 mL (1 mg/mL) was mixed with 100 mL of the prepared nanogold particles. The mixture was lightly shaken for 15 min. The blocking was carried out by adding polyethene glycol (PEG-20,000) 1% m/v final concentration with gentle stirring for another 15 min, followed by centrifugation at 12,000 rpm for one hour. The conjugated nanogold particles were suspended in 1 mL dilution buffer containing (sucrose3% (w/v), 20 mM Tris, bovine serum albumin 1% (w/v), and sodium azide 0.02% (w/v)) and stored at 4 °C [18].

### 2.7. Dispensing of the Prepared SARS-CoV-2 NP-Specific Rabbit IgG Conjugated Nanogold Particles and Non-Conjugated SARS-CoV-2 Spike Protein Specific Goat IgG in the Nitrocellulose Membrane and the Conjugation Pad

**The sample pad:** Glass fiber (Ahlstrom 222), pretreated with sample pad buffer solution pH 8.5 (ultrapure water containing 1% (w/v) PVP, 2% (w/v) titronX100, 3.81% (w/v) borax 0.1% (w/v), 0.15% (w/v) SDS, casein sodium salt, 0.5% (w/v) sodium cholate and 0.02% (w/v) sodium azide) was dried at 37 °C. [19], as shown in Figure 1.

**The conjugation pad:** Glass fiber (Ahlstrom 8964), pretreated with conjugation treated buffer solution pH 7.4 (20 mM PBS contained 2% (w/v) BSA, 2.5% (w/v) sucrose, 0.3% (w/v) PVP, 1%(w/v) triton x100, and 0.02% (w/v) sodium azide), was dried at 37 ℃. Then, it was saturated with SARS-CoV-2 NP-specific rabbit IgG conjugated nanogold particles; it was dried at 37 ℃ for one hour and kept in dry conditions.

**Nitrocellulose (NC) membrane (mdi CNPF-PD31):** The dispenser (Iso-flow) was used to dispense two lines on the NC membrane (300 mm × 25 mm). The purified goat IgG specific to SARS-CoV-2 spike protein (1.2 mg/1 mL) was dispensed at the test line (1 µL/1 cm line) and the goat anti-rabbit IgG antibodies (Jackson immune-research laboratories USA) (0.6 mg/mL) was dispensed at the control line (1 µL/1 cm line). The loaded NC membrane was dried at 37 °C for four hours and kept in dry conditions. The PVC card was used to fix the treated sample pad, conjugated pad, loaded NC and absorbent pad as on strip, then cut in 4 mm width, as shown in Figure 1.

### 2.8. Specificity Testing Using Other Viral Strains

The developed LFI-COVID-19 antigen test was examined with different viral strains, including adenovirus type 3/human coronavirus 229e/influenza A H1N1/human rhinovirus 2/parainfluenza virus 2 and respiratory syncytial virus in a titer of 10^6^ TCID_50_/mL for each strain.

### 2.9. Cross-Reactivity and Interfering Substances

Using the developed LFI COVID-19 antigen test, any possible cross reaction was investigated using different bacterial strains (10^7^ CFU/0.1 mL), including *Candida albicans, Escherichia coli*, *Staphylococcus aureus*, *Corynebacterium*, *Pseudomonas aeruginosa*, *Staphylococcus epidermidis*, *Streptococcus pneumonia* and *Streptococcus pyogenes*. In addition, the interfering substances, such as whole blood 20 μL/0.1 mL, dexamethasone 10 mg/mL, Tamiflu 10 μg/mL, phenylephrine 10 mg/mL, were tested using the LFI COVID-19 antigen test.

### 2.10. Sensitivity Testing of Developed LFI-COVID-19 Antigen

The limit of detection or the minimal amount of the virus that can be detected using the developed LFI-COVID-19 antigen test was determined as follow: tenfold serial dilutions starting from 10^8^ TCID_50_/0.1 mL to 10 TCID_50_/0.1 mL were tested with the developed test and using the RT-qPCR commercial kit (TRAN. DV101, China).

### 2.11. Determination of Sensitivity, Specificity and Accuracy of the Prepared LFI COVID-19 Antigen as Compared with RT-qPCR Commercial Kit 

Two hundred specimens (swabs) that were collected from COVID-19 suspected patients were examined for SARS-CoV-2 virus detection with the developed LFI-COVID-10 antigen test and using RT-qPCR as the gold standard test method. The samples were considered positive if RT-qPCR indicated a positive result. Specimens were considered negative if RT-qPCR indicated a negative result [14,15].

### 2.12. Ethical Approval

The Institutional Animal Care and Use Committee at the Central Laboratory for Evaluation of Veterinary Biologics hereby acknowledges the research manuscript and it has been reviewed under our research authority and is deemed compliant with bioethical standards in good faith.

## 3. Results

### 3.1. Gold Nanoparticle Measuring

The gold solution reached a peak of 529.02, which indicated that the particle size was 40 nm (Figure 2).

### 3.2. Sensitivity Testing

The minimal virus titer (TCID_50_/0.1 mL) that the LFI-COVID-19 antigen test could detect was 10^3^ TCID_50_/0.1 mL, which was weak or suspected positive but the virus titer 10^4^ TCID_50_/0.1 ml was definitely positive, as indicated in Figure 3. Meanwhile, the sensitivity testing was confirmed using the real time RT-PCR technique; the results indicated that the LFI-COVID-19 antigen test was positive at the Ct value of 31.1 and weak positive at the Ct value 33.4, as shown in Table 1.

### 3.3. Specificity Testing

The LFI-COVID-19 antigen test was positive with SARS-CoV-2 antigens and negative for all other tested virus species, including adenovirus type 3, human coronavirus 229e, influenza A H1N1, human rhinovirus 2, parainfluenza virus 2, and respiratory syncytial virus.

### 3.4. Cross-Reactivity and Interfering Substances

All the interfering substances and tested bacterial strains indicated negative results after testing with the developed LFI-COVID-19 antigen test, while the tested SARS-CoV-2 antigens indicated positive results and no substances showed any interference with the LFI-COVID-19 antigen test.

### 3.5. The Relative Sensitivity, Specificity and Accuracy of LFI COVID-19 Antigen Test Compared with RT-qPCR Commercial Kit

The sensitivity, specificity and accuracy for the developed LFI-COVID-19 antigen test relative to RT-qPCR were 95%, 98%, and 97%, respectively, as shown in Table 2.

## 4. Discussion

SARS-CoV-2 is the seventh most infectious coronavirus of homo-sapiens and has a higher transmissibility rate than the previous coronaviruses [20]. This virus also causes mild to severe respiratory system illness. Moreover, it spreads to other organs and systems, including but not limited to the liver, testes, GI, and CNS [21].

The COVID-19 pandemic has created an unaccustomed demand for RT-qPCR testing all over the world. Direct SARS-CoV-2 virus antigen-based techniques, such as LFA, are an alternative method to the RT-qPCR technique and are likely to be the only viable, rapid and less expensive solution for low- and middle-income countries (LMICs), such as in Egypt. In the current study, we developed the LFI-COVID-19 antigen test for rapid diagnosis of COVID-19 virus infection through the detection of SARS-CoV-2 nucleocapsid and spike proteins in clinical specimens. SARS-CoV-2 nucleocapsid protein-specific IgG prepared in rabbits and conjugated with gold-nanoparticles, in addition to goat IgG specific to SARS-CoV-2 spike antigens, have been used for developing SARS-CoV-2 antigen detection LFI [22]. Interestingly, many producers of LFI rapids tests, including LabCorp, Abbott, and Quest Diagnostics, reported no change in the accuracy of diagnostics towards the current variants [23].

Evaluation of the sensitivity of the developed LFI-COVID-19 antigen test indicated that the limit of detection (LOD) of SARS-CoV-2 NP and spike proteins was 10^3^ TCID_50_/0.1 mL, which gave a weak positive result. A virus titer of 10^4^ TCID_50_/0.1 mL, however, was clearly positive. When the obtained results were correlated with the gold standard RT-qPCR testing, the LFI-COVID-19 antigen test was positive at the Ct value 31.1 and weak positive at Ct value 33.4. The WHO has proposed an optimal value of 10^4^ genomic copies/mL and a minimal LOD of 10^6^ genomic copies/mL [10]; thus, the LOD of the developed test, as tested, is within the range of these guidelines.

To validate the specificity of the developed LFI-COVID-19 antigen test, it was tested for the detection of SARS-CoV-2 and other different viruses (adenovirus type 3, human coronavirus 229e, influenza A H1N1, human rhinovirus 2, parainfluenza virus 2, and respiratory syncytial virus). All tested viruses indicated negative results, except SARS-CoV-2. Furthermore, the prepared LFI-COVID-19 antigen test indicated no interference with any substances, such as whole blood 20 μL/0.1 mL, dexamethasone 10 mg/mL, Tamiflu 10 μg/mL and phenylephrine 10 mg/mL, or any different bacterial strains of 10^7^ CFU/0.1 mL, including *Pseudomonas aeruginosa*, *Candida albicans*, *Staphylococcus aureus*, *Corynebacterium* spp., *Staphylococcus epidermidis*, *Escherichia coli*, *Streptococcus pneumonia*, and *Streptococcus pyogenes*). In similar studies, it was reported that the LFI for the detection of SARS-CoV-2 NP was evaluated for specificity validation using nasopharyngeal samples containing coronaviruses 229E, OC43, HKU1, and NL63, as well as SARS-CoV-2. It was observed that SARS-CoV-2 indicated positive results, while the other viruses indicated negative results [24]. In another investigation, LFA based on SARS-CoV-2 nucleoprotein monoclonal antibodies was developed, and it did not show cross-reactivity with human coronaviruses, influenza A, influenza B, chicken IBV, MERS-CoV N protein, rubella virus, or pyogenic bacteria; however, it cannot distinguish between SARS-CoV and SARS-CoV-2 [25].

The performance of the developed LFI-COVID-19 antigen test was evaluated by comparing it with the RT-qPCR as a master test. The developed test exhibited 95%, 98%, and 97% for sensitivity, specificity, and accuracy, respectively. This complies with the WHO TTP, which proposed an acceptable specificity and sensitivity target of 97 and 80%, respectively, relative to RT-qPCR, while the desired specificity and sensitivity target is 99 and 90%, respectively, relative to RT-qPCR [10]. Interestingly, in previous studies, the LFA for the detection of SARS-CoV-2 (NP) indicated a specificity and sensitivity of 99.5 and 57.6%, respectively [25]; in another study, it was reported that the BinaxNOW™ Rapid Antigen Test for SARS-CoV-2 indicated 82 and 100% for sensitivity, and specificity, respectively, relative to RT-qPCR [26]. In addition, the QuickVue SARS Antigen Test indicated 96.6% and 99.3% for sensitivity and specificity, respectively. On the other hand, the COVID-19 Home Test (Ellume) indicated 95 and 97% for sensitivity, and specificity, respectively [27]. The FDA estimated that antigen LFA tests’ performance varies across a wide range between 90.0 and 100% for sensitivity [28]. The FDA addressed the high rates of false positives for antigen LFA tests [29] and claims sensitivities of 90% or higher to be optimal; however, the real-world performances indicated lower sensitivity values than originally claimed, as shown in many independent studies for LFAs (refs. [30,31]). The sensitivity deviations from preclinical data could be attributed to some factors, such as non-random sampling, user error, and disease prevalence (refs. [32,33]). The developed LFI-COVID-19 antigen test is a point of care and an alternative approach to current laboratory methods, especially RT-qPCR, providing cheap, rapid, easy, and on-site COVID-19 diagnosis, if it is manufactured on a large scale for commercial use.

## 5. Conclusions

The developed LFI-COVID-19 antigen test is an alternative point of care approach to the current laboratory methods, especially RT-qPCR. It provides an easy, rapid, and on-site diagnostic tool for COVID-19 infection, and it is a cheap test if it is manufactured on a large scale for commercial use.

## 6. Limitations

The performance of the developed LFI-COVID-19 Ag test was probably affected by different issues, including user error, sampling method, and number of samples, which need further investigations.

## Figures and Tables

**Figure 1 nanomaterials-12-02477-f001:**
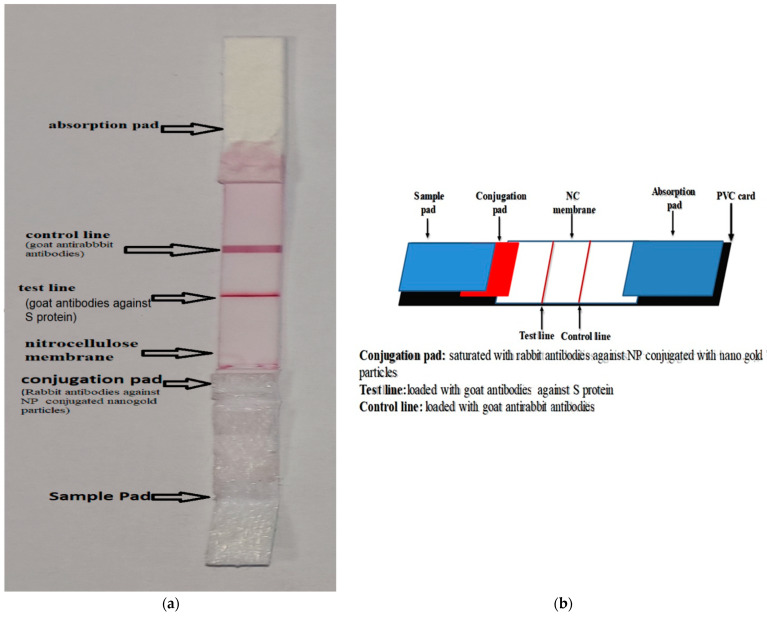
(**a**) The lateral flow assay (LFA) description using the prepared LFA strip showed sample pad, conjugation pad, nitrocellulose membrane, test line, control line and absorption pad; (**b**) lateral flow assay schematic representation of the SARS-CoV-2 virus/antigen visual readout. The virus/antigen is loaded on the sample pad of the LFA strip. The rabbit anti NP antibodies conjugated nanogold particles bind with SARS-CoV-2 virus/antigen and the complex flows down the LFA strip. The test line (T) could be visualized only when SARS-CoV-2 virus/antigen detected by rabbit anti NP antibodies conjugated nanogold particles, whereas the control line (C) should appear for the test validity.

**Figure 2 nanomaterials-12-02477-f002:**
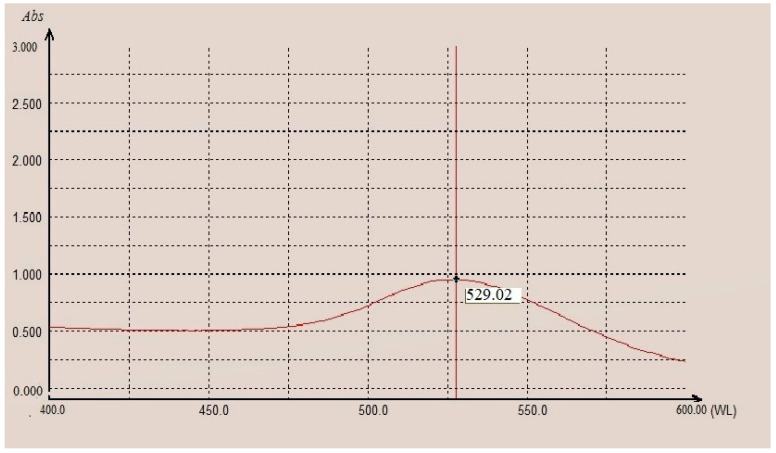
Spectrophotometer curve for 40 nm colloidal gold nanoparticles.

**Figure 3 nanomaterials-12-02477-f003:**
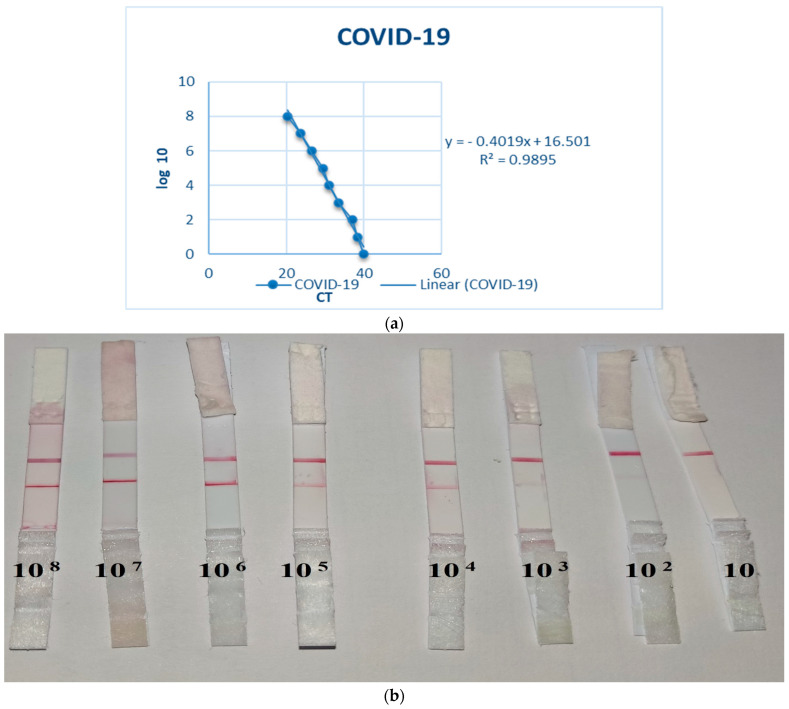
(**a**) Linear range and the calculation equation for SARS-CoV-2; (**b**) Limit of detection (LOD) for SARS-CoV-2 detection by the developed LFI for antigen detection using serial dilutions of phosphate buffer saline (PBS) spiked with SARS-CoV-2.

**Table 1 nanomaterials-12-02477-t001:** Limit of detection (LOD) of SARS-CoV-2 antigens by the developed LFI antigen test, as compared to RT-qPCR using phosphate buffer saline samples spiked with SARS-CoV-2.

Method	10^8^	10^7^	10^6^	10^5^	10^4^	10^3^	10^2^	10
LFI COVID-19 antigen	+ve	+ve	+ve	+ve	+ve	Weak positive	−ve	−ve
RT-qPCR	+ve (Ct20.3)	+ve (Ct23.7)	+ve (Ct26.5)	+ve (Ct29.5)	+ve (Ct31.1)	+ve (Ct33.4)	+ve (Ct37.1)	+ve (38.3) *

* According to the insert pamphlet, when the Ct- cycle threshold values of a target gene are between 38 and 40, the sample should be regarded as suspected negative (−ve).

**Table 2 nanomaterials-12-02477-t002:** The relative sensitivity, specificity and accuracy of the developed LFI COVID-19 antigen test, as compared with the RT-PCR commercial kit.

A *				
Method	RT-qPCR			Total Results
LFI COVID-19 antigen	Results	Positive	Negative	
	positive	35 (True +ve)	3 (False +ve)	
	Negative	2 (False −ve)	160 (True −ve)	
Total results		37	163	200
**B**				
Sample	Sensitivity (%)	Specificity (%)	Accuracy (%)	
SARS-CoV-2 virus	95%	98%	97%	

* (**A**) True positive (LFI+ PCR+); false positive (LFI+ PCR−); true negative (LFI− PCR−); false negative (LFI− PCR+); (**B**) the clinical performance of developed LFI-COVID-19 Ag: sensitivity, specificity and accuracy.

## Data Availability

Not applicable.

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
