# Peer review of "Development of Lateral Flow Immunochromatographic Test for Rapid Detection of SARS-CoV-2 Virus Antigens in Clinical Specimens"

_nanomaterials, 2022, doi:10.3390/nano12142477_

Round 1

Reviewer 1 Report

My major concerns were addressed by the authors, except a review by an English speaking editor is still needed to revise the grammar.  

In the course of reading this manuscript again, I found additional items that should be corrected.  In Table 2, the contents of the table do not match the contents of the text in the caption, specifically related to the results that are False Positive and False Negative.  I believe the PCR is the "gold standard" in this table, so when the PCR is positive and lateral flow is negative, this should be False Negative.  The caption describes this properly, but the table appears to mix up the false positives and false negatives.  Also, in lines 149 and 150, the use of decimal points in the denominator of concentrations is very unusual, in my opinion.  I think it is more conventional to use "mL" in the denominator, rather than "0.1 mL"; multiplying the numerator and denominator by 10 should result in the same concentration expressed in more conventional format.

Author Response

Response to reviewer 1#

  1. In Table 2, the contents of the table do not match the contents of the text in the caption, specifically related to the results that are False Positive and False Negative I believe the PCR is the "gold standard" in this table, so when the PCR is positive and lateral flow is negative, this should be False Negative. The caption describes this properly, but the table appears to mix up the false positives and false negatives.

Response: the error has been revised page 7 . table 2 . yellow highlights

  1. Also, in lines 149 and 150, the use of decimal points in the denominator of concentrations is very unusual, in my opinion. I think it is more conventional to use "mL" in the denominator, rather than "0.1 mL"; multiplying the numerator and denominator by 10 should result in the same concentration expressed in more conventional format.

Response:  have been corrected page 4 . in lines 149, 150, 238 and 239 yellow highlights

Reviewer 2 Report

The authors have carefully revised the article based on the comments of the reviewers. I think it has now reached the level of publication in Nanomaterials.

Author Response

Thanks for your valuable comments and your efforts

Reviewer 3 Report

This paper illustrates the lateral flow immunoassays (LFIA) for detecting SARS-CoV-2 antigen based on gold nanoparticles. Au is usually used as a standard material for fabricating LFIA. The creativity is not good enough, but the whole research is worth to be published. Specific comments are listed below. 

1. The second author institute missed the country information. 

2. The conjugate pad and absorbent pad are not stable on the backing card, why?

3. Why are lots of yellow and green marks in this manuscript? 

4. It is better not to name “Nanogold Lateral Flow Immunoassay-based technique” as “LFI-COVID Ag”, no need to add the initial antigen as “Ag” here. Since the authors used gold here, “Ag” also means sliver. Readers are easily misunderstood here.

The detection liner range and LOD should be given as a Figure.

5. Even though the authors used a typical method to synthesize this nanogold, it still needs to provide some characterization of materials.

Author Response

Response to reviewer 3#

  1. The second author institute missed the country information.

Response: has been edited page 1 . in line 8, yellow highlights

  1. The conjugate pad and absorbent pad are not stable on the backing card, why?

Response: actually they are stable but they are probably seems to be not stable in figure 1 but once it is fixed in plastic cassette, they would be more stable

  1. Why are lots of yellow and green marks in this manuscript?

Response: they were for reviewers and editor responses, but they are replaced by recent changes.

  1. It is better not to name “Nanogold Lateral Flow Immunoassay-based technique” as “LFI-COVID Ag”, no need to add the initial antigen as “Ag” here. Since the authors used gold here, “Ag” also means sliver. Readers are easily misunderstood here.

Response: LFI-COVID Ag has been replaced by LFI-COVID Antigen in the whole manuscript, please notice the yellow highlights

The detection liner range and LOD should be given as a Figure

Response: it was edited page 6 figure 3

  1. Even though the authors used a typical method to synthesize this nanogold, it still needs to provide some characterization of materials.

Response :  you are absolutely right, but our institute is focusing on biologics, vaccines and diagnostics so we don’t have these facility to conduct these characterization and we are used to develop nanogold particles on standardized SOP using gold chloride which was supplied from sigma, and this standardized SOP could be confirmed with spectrophotometer reading values. We used this approach in different previous researches and it worked well with us. Thanks for your valuable comments    

Round 2

Reviewer 3 Report

The author made good revisions, no questions anymore.

This manuscript is a resubmission of an earlier submission. The following is a list of the peer review reports and author responses from that submission.

Round 1

Reviewer 1 Report

In this manuscript, authors developed Nanogold Lateral Flow Immunoassay-based technique (LFI-COVID Ag) for the detection of SARS-CoV-2 nucleocapsid protein. The results show that this method has excellent performance parameters such as cheap, easy and rapid, and is an alternative method approach to the current laboratory methods especially RT-qPCR. Perhaps this is a viable and rapid solution for low- and middle-income countries. Obviously, this has important practical application value. However, the use of AuNPs as a visible light indicator for Lateral Flow Immunoassay analysis, that is, the basic technology for pregnancy testing has been very mature and commercialized, so this study is slightly insufficient in terms of theoretical innovation.

Suggestions for revision are as follows:

1. There are some formatting errors in the text, and the chemical formula needs to be written correctly. For example Section 2.6 “K2CO3”, Section 2.5 “HAuCl4”. Atomic numbers should be subscripted should be subscripted.

2. The figures in the text are not clear, please increase the resolution, such as Fig. 1 and Fig. 2.

Reviewer 2 Report

Some sections are written well in what I would call "proper English," but other sections require significant editing, particularly in grammar and punctuation.  There is also inconsistent use of capitalization and italics, in text and in figures/tables.  The text used in labelling parts of Figure 1 is mostly not readable (please change the font and/or font size).

The methods are described fairly well, except it is not clear why the authors consider the antibodies they generated as "monospecific."  Several questions come to mind:

1) Is "monospecific" the same as "monoclonal"?  What is the difference?

2) If agar gel precipitation is important in establishing the "monospecificity", these experiments should be described in a little more detail.

3) If "monospecificity" is based on something else (e.g. specificity studies), that should be pointed out.

While it is great that the authors could develop this test on their own in a LMIC, the resources used in this work (especially the animals) are a significant resource, and this approach is still probably not practical in most LMIC settings.  Access to kit materials and commercially available purified antigens is also important.  Isn't it still less costly to buy validated commercial kits?  Perhaps an analysis of the relative costs (make vs. buy) would be of interest to many readers.